# Application of Transcriptome Analysis to Understand the Adverse Effects of Hypotonic Stress on Different Development Stages in the Giant Freshwater Prawn *Macrobrachium rosenbergii* Post-Larvae

**DOI:** 10.3390/antiox11030440

**Published:** 2022-02-22

**Authors:** Bo Liu, Qiang Gao, Bo Liu, Changyou Song, Cunxin Sun, Mingyang Liu, Xin Liu, Yunke Liu, Zhengzhong Li, Qunlan Zhou, Hao Zhu

**Affiliations:** 1Wuxi Fisheries College, Nanjing Agricultural University, Wuxi 214081, China; liubo@njau.edu.cn (B.L.); 2017113009@njau.edu.cn (M.L.); sdhzlb1@163.com (X.L.); 2015213006@njau.edu.cn (Y.L.); zhouql@ffrc.cn (Q.Z.); 2Key Laboratory of Aquatic Animal Nutrition and Health, Freshwater Fisheries Research Center, Chinese Academy of Fishery Science, Wuxi 214081, China; songchangyou@ffrc.cn (C.S.); suncx@ffrc.cn (C.S.); 3Zhejiang Institute of Freshwater Fishery, Huzhou 313001, China; swausc@163.com; 4College of Fisheries and Life Science, Shanghai Ocean University, Shanghai 201306, China; 2013813026@njau.edu.cn; 5Fishery Machinery and Instrument Research Institute, Chinese Academy of Fishery Sciences, Yangpu District, Shanghai 201306, China; zhuhao0511@163.com

**Keywords:** *Macrobrachium rosenbergii*, hypotonic stress, development stages, transcriptome analysis, antioxidant capacity

## Abstract

Salinity is one of the important environmental factors affecting survival and growth of aquatic animals. However, the impact of low-salinity stress on *M. rosenbergii* post-larvae at different development stages remains elusive. Therefore, the aim of this study was to explore the underlying mechanisms of hypotonic stress at different development stages of *M. rosenbergii* post-larvae through transcriptome analysis and antioxidant parameters detection. The salinity of the control group was 15 psu (S15) and the hypotonic stress group was 6 psu (S6). Samples were collected at 7 days-post-hatch (dph), 14 dph and 21 dph larvae. The results showed that hypotonic stress caused oxidative damage in post-larvae evidenced by decreased glutathione peroxidase (GSH-Px); superoxide dismutase (SOD); anti-superoxide anion free radical (ASAFR); and increased malondialdehyde (MDA); nitric oxide (NO); and inducible nitric oxide synthase (iNOS) levels. Transcriptome analysis showed that there were 1428, 1187, 132 DEGs including 301, 366, 4 up-regulated genes and 1127, 821, 128 down-regulated genes at 7 dph, 14 dph and 21 dph larvae under hypotonic stress, respectively. Furthermore, GO and KEGG enrichment indicated that hypotonic stress led to dysregulation of immune signals including lysosome and autophagy in the 7 dph larvae. The autophagy-related genes including beclin 1-associated autophagy-related key regulator (*Barkor*); ubiquitin-like modifier-activating enzyme ATG7 (*ATG7*); *Beclin*; autophagy-related protein 13 (*ATG13*); nuclear receptor-binding factor 2 (*Nrbf2*); ubiquitin-like-conjugating enzyme ATG3 (*ATG3*); vacuole membrane protein 1 (*VMP1*); and autophagy-related protein 2 (*ATG2*) decreased at 7 dph, and 14 dph larvae, and then increased at 21 dph larvae under hypotonic stress. In the 14 dph and 21 dph larvae, the renin-angiotensin system was activated. In conclusion, our data indicated that hypotonic stress reduced the antioxidant capacity and impaired the immune system in post-larvae, but as development progresses, the adaptability of post-larvae to hypotonic stress gradually increased, and might reach a new homeostasis through the RAS signaling pathway.

## 1. Introduction

Salinity is a vital environment factor that affects survival, growth, and distribution of many aquatic organisms and mainly affects the physiology of various aquatic animals [1,2]. Under salinity fluctuations, some of the aquatic animals required more energy to adjust the osmotic pressure balance between their body fluid and the environment [3,4]. A series of studies showed that salinity affected the feeding behavior, growth, and health of *Macrobrachium nipponense* [5], *Litopenaeus vannamei* [6], and *Nephrops norvegicus* [7], indicating that water salinity is an important factor in the growth and survival of crustacean species. The way in which aquatic animals respond to ambient salinity stress is quite complex and extends from behavior to the molecular level [8]. To achieve ion homeostasis during salinity stress, aquatic animals must activate appropriate signal transduction pathways and send messages to specific target molecules by signal transducers to restore homeostasis [9,10]. However, information about the signal transduction events for osmoregulation in crustacean species is limited.

*Macrobrachium rosenbergii*, also termed as the giant freshwater prawn, is known worldwide for its excellent growth performance, high market demand, high resistance to stress, and adaptation to euryhaline environments [11,12]. This species lives in tropical and subtropical freshwater environments with access to adjacent brackish water areas, and high concentrations of environmental salinity was required during post-larval development stages [13]. The post-larval stages need about 21 days and then transfer into freshwaters for final growth and maturation into adults. The environmental salinity plays a vital role in the development of *M. rosenbergii* embryo and post-larvae [14]. Previous studies showed that *M. rosenbergii* could tolerate 0–18 psu salinity during the entire life cycle [15], and the optimal developmental salinity is 15 psu in the post-larvae stage of *M. rosenbergii* [16]. However, it is difficult to build a large aquatic nursery farm in coastal waters, and large amounts of sea salt were needed for seeding in inland areas, which leads to rising production costs. In addition, the seedling-tail water discharge will cause land salinization and impose adverse effects on the environment, which is against the requirements of sustainable development. Accordingly, exploring effective methods to reduce the salinity of the seedling water has become an urgent scientific problem in the breeding of *M. rosenbergii*. First of all, it is essential to recognize the effect of low salinity stress on the post-larval stage of *M. rosenbergii*. However, the information required on osmoregulation in relation to salinity stress in *M. rosenbergii* post-larvae is not currently available. The illumination of the mechanisms for *M. rosenbergii’s* adaptation to salinity fluctuations would improve the *M. rosenbergii* breeding culture.

On the other hand, the ability to osmoregulation is related to size, nutritional conditions, and developmental stages [17]. Previous study showed that the survival of 2 g *L. vannamei* juveniles at 5 practical salinity units (psu) or 40 psu exhibited no significant difference [18]. Another study showed that *L. vannamei* post-larvae and juveniles could be grown successfully at 4 psu [19]. However, juveniles were able to survive as well at 2 psu as at 30 psu, while salinities below 4 psu put an adverse effect on post-larvae survival and growth. According to the above studies, the different developmental stages of *L. vannamei* showed different levels of adaptability to low salinity. However, little research has been performed on adaptability to low salinity conditions at different developmental stages of *M. rosenbergii* post-larvae.

In this study, antioxidant parameters and RNA-Seq were used to study the responses of *M. rosenbergii* post-larvae under hypotonic stress to evaluate the adaptability of osmoregulatory mechanisms at different development stages of *M. rosenbergii* post-larvae. As far as we know, this is the first report on the adaptation mechanism of *M. rosenbergii* post-larvae to low salinity stress at different development stages. The results will provide in-depth knowledge of physiological mechanisms at different development stages of *M. rosenbergii* post-larvae under low salinity stress.

## 2. Materials and Methods

### 2.1. Ethics Statement

This research received permission from the Animal Care and Use Committee of Nanjing Agricultural University (Nanjing, China). All animal procedures were performed according to the Guidelines for the Care and Use of Laboratory Animals in China.

### 2.2. Experimental Animal, Design and Sampling

The *M. rosenbergii* used in the present experiment were obtained from Zhejiang Aquatic Seed Industry Co., Ltd., Huzhou City, in China. Parental prawns with excellent traits were selected, and the weight of females and males was about 30–50 g, respectively. The ratio of male-to-female was 3:1. The water temperature was between 28 and 30 °C, and the water quality was maintained as follows: pH 7.5–8.4, NH_3_ < 0.05 mg/L, DO > 6 mg/L. Prawns were fed with commercial feed (crude protein: 41%, crude lipid 8%), 3 times a day to achieve a significant satiety (06:30, 11:30, 17:30). *M.rosenbergii* required about 28 days to complete the embryonic development process. When the embryos ruptured into post-larvae, we removed the post-larvae into another concrete pond (2.1 × 1.5 × 0.65 m). There were 60,000 post-larvae of uniform size randomly distributed into two groups (6 tanks, 500 L water/tank, 3 tanks/group, and 10,000 individuals/tank) for the salinity stress experiments. The salinity of the control group was 15 psu (S15) and the hypotonic stress group was 6 psu (S6), and farming lasted 21 days. We continuously inflated and increased the dissolved oxygen day and night to ensure sufficient oxygen; reduced human interference; reduced noise levels; and prevented additional stress. The water quality was maintained as follows: temperature 30 ± 0.5 °C, pH 7.7–8.8, NH_3_ < 0.05 mg/L, DO > 8.6 mg/L. The 0 and 7 day-post-hatch (dph) larvae were fed with artemia, with egg custard added on day 8, 4 times daily (07:00, 11:00, 15:00, 19:00).

During the farming days, 9 post-larvae samples (about 2–3 g /tank, 3 tanks/group) in the stage of 7 dph (S15-7; S6-7), 14 dph (S15-14; S6-14), 21 dph larvae (S15-21; S6-21) from the control and hypotonic stress group were randomly collected, respectively, as shown in Appendix A (Appendix A). Then samples were rapidly frozen in liquid nitrogen and stored at −80 °C for later parameters’ measurement and RNA extraction.

### 2.3. Antioxidant Parameters Assays

Homogenate post-larvae samples in ice-cold physiological saline, centrifuged at 3000 rpm for 10 min at 4 °C, and the supernatant was preserved for antioxidant parameters analysis. According to the method, the levels of superoxide dismutase (SOD) of nine samples in each group were measured by spectrophotometric means using devices from Bio-tek Instruments, Winooski, VT, Inc., USA [20]. Glutathione peroxidase (GSH-Px) and anti-superoxide anion free radical (ASAFR) of nine samples in each group were determined with colorimetric method, and malondialdehyde (MDA) was determined with TBA method [21]. The nitric oxide (NO) and inducible nitric oxide synthase (iNOS) of nine samples in each group were determined by the nitrate reductase method and colormetric method [22]. The test kit for the tests was acquired from the Nanjing Jiancheng Biological Engineering Research Institute.

### 2.4. RNA Extraction, cDNA Library Construction and Sequencing

There were two different salinity levels (6 psu, 15 psu) and three time points (7 dph, 14 dph, 21 dph). Each group had three biological replicates, so that eighteen samples were used to extract RNA. RNA extraction was following methods by Chen[23]. RNA was extracted from each sample using RNAiso Plus reagent (Dalian Takara Co. Ltd., China), and measured by spectrophotometer. Quantity and quality of the RNA was assessed by OD260/280 method and electrophoresis in 1% agarose gel. The cDNA was synthesized by purified RNA and repaired to construct a sequencing library. The libraries were then sequenced on the Hiseq PE150 platform, and produced 150 bp paired-end reads.

### 2.5. Transcriptome Data Analysis

In order to get quality data, the first step was to quality evaluate and filter the original data. Using FastQC (http://www.bioinformatics.babraham.ac.uk/projects/fastqc/ (accessed on 25 January 2022)) to quality control the sequencing data, and using SOAPnuke software (https://github.com/BGI-flexlab/SOAPnuke/ (accessed on 25 January 2022)) to filter the raw reads based on three requirements: adapter reads were removed; unknown nucleotides (>10%) reads were removed; low quality reads (Q-value < 20) were removed.

Qualified reads after filtering were assembled with Trinity software (http://trinityrnaseq.github.io/ (accessed on 25 January 2022)) for de novo. The transcriptome sequence assembled by Trinity was used as the reference sequence, and the RSEM software was used to call the Bowtie2 program to align the clean reads of each sample to the reference sequence. The FPKM (fragments per kilo base of exon per million fragments mapped) was used to indicate the gene expression level. The abundance of gene expression was expressed as the logarithm of the fold change. In order to obtain comprehensive gene function information, BLAST software was used to compare the transcript assembly sequence with NCBI protein sequence database (NR), SwissProt protein sequence database, Pfam, KOG, and GO databases to obtain the annotation information.

The edgeR software package was used to analyze the differentially expressed genes (DEGs) between two groups, and the gene expression with FDR (false discovery rate) < 0.05 and |log FC| (fold change) ≥ 1 was considered to be significant difference. Besides, in order to identify the main biological functions of DEGs, the gene ontology (GO) enrichment analysis was used with Blast2GO and OmicShare. Based on the Blast2GO function annotation, the KEGG pathway enrichment analysis was performed on DEGs.

### 2.6. Quantitative Real-Time RT-PCR (qPCR) Validation

RNA was extracted from each sample using RNAiso Plus reagent (Dalian Takara Co. Ltd., China), and measured by spectrophotometer. The quantity and quality of the RNA was assessed by OD260/280 method and electrophoresis in 1% agarose gel. After normalizing the concentration of the RNA samples, cDNA was generated from 500 ng DNase-treated RNA using ExScriptTM RT-PCR kit according to the manufacturer’s directions (Takara Co., Ltd., Tokyo, Japan). The cDNA synthesis was performed according to the following steps: 4 μL 5× gDNA Eraser Buffer, 2 μL gDNA Eraser, 1 μL RNA, 13 μL RNase free dH_2_O were incubated at 42 °C for 2 min, and then added 2 μL PrimerScriptRT Enzyme mix 1, 2 μL RT Primer Mix, 8 μL 5× Primerscript Buffer 2(for Real Time), 8 μL RNase free dH_2_O in 37 °C for 15 min, 85 °C for 5 s. Primers for each gene were designed by primer 5.0 based on the sequences obtained from this transcriptome sequencing library (Appendix A). All primers were synthesized by Shanghai Generay Biotechnology, Co., Ltd., China. Real-time quantitative PCR (RT-PCR) was performed with SYBR Green II Fluorescence Kit (Takara Co., Ltd., Japan) using a real-time quantitative detector (BIO-RAD, United States) according to the method reported by Sun [24]. The β-actin was selected as the housekeeping gene to normalize our samples because of its stable expression in the present study. The qPCR reactions were carried out in triplicate of each sample and further calculated using the 2^−ΔΔ^ method [25].

### 2.7. Correlation Analysis

According to the method of Liu [26], Pearson’s correlation test was performed to find the correlations between antioxidant parameters and key different expression genes in hypotonic stress group. The significance threshold was set at a *p*-value of less than 0.05. The heatmap was created in R with the pheatmap package following the method reported by Zhao et al. (2017) [27].

### 2.8. Statistical Analysis

For the antioxidant parameters, data were analyzed by two-way analysis of variance (ANOVA) using SPSS 20.0 to detect the independent and interactive effects of salinity and development stage. Significant differences in the groups (*p* < 0.05) were estimated by multiple range test of Duncan’s. One-way analysis of variance was used to analyze all the data in one group if the interaction between the salinity and development stage was significant.

## 3. Results

### 3.1. Effects of Hypotonic Stress on Antioxidant Parameters of M. rosenbergii Post-Larvae at Different Development Stages

Glutathione peroxidase activity (GSH-Px), superoxide dismutase activity (SOD), anti-superoxide anion free radical compounds (ASAFR), and malondialdehyde (MDA) levels of different development stages in *M. rosenbergii* post-larvae under hypotonic stress are shown in Figure 1A–D. Significant (*p* < 0.05) interactions of hypotonic stress and development stages on SOD and MDA were observed. Besides, all of the parameters were significantly affected by salinity and development stages. At 7 dph, 14 dph and 21 dph larvae, GSH-Px activity and ASAFR were decreased significantly in the S6 group compared with those of the S15 group (*p* < 0.05). At 14 dph and 21 dph larvae, the SOD activity in the S6 group was significantly lower than that of the S15 group (*p* < 0.05), while they showed no significant differences at 7 dph larvae (*p* > 0.05). At 7 dph larvae, the MDA level in the S6 group was significantly increased compared to that of the S15 group (*p* < 0.05), while it showed no significant differences at 14 dph and 21 dph larvae (*p* > 0.05). Besides, with the development stages from 7 dph to 21 dph, GSH-Px activity, SOD activity, and ASAFR were increased significantly in the control (S15) and hypotonic stress (S6) groups (*p* < 0.05). The MDA level was decreased significantly in the S15 and S6 groups from 7 dph to 21 dph larvae (*p* < 0.05).

### 3.2. Effects of Hypotonic Stress on NO and iNOS of M. rosenbergii Post-Larvae at Different Development Stages

Nitric oxide (NO) and inducible nitric oxide synthase (iNOS) level of different development stages in *M. rosenbergii* post-larvae under hypotonic stress are shown in Figure 1E,F. Significant (*p* < 0.05) interactions of hypotonic stress and development stages on iNOS were observed. Moreover, the iNOS level was significantly affected by development stages, and both NO and iNOS were affected by salinity. At 7 dph, 14 dph and 21 dph larvae, the levels of NO and iNOS were increased significantly in the S6 group compared with those of the S15 group (*p* < 0.05). Besides, with the development stages from 7 dph to 21 dph, the iNOS level was increased and then decreased significantly with the developmental stages (*p* < 0.05). The NO level showed no significant difference between the different stages in the same group (*p* > 0.05).

### 3.3. The Quality of Library Sequencing 

As shown in Appendix A (Appendix A), RNA-Seq generated more than 34,119,258 raw reads for each library, with an average of 49,419,690; 46,172,288; 40,117,424; 55,110,425; 53,303,988; 41,847,539 paired-end reads for the S15-7; S15-14; S15-21; S6-7; S6-14; S6-21 groups, respectively. The clean ratio of raw reads generated more than 78.85%. The GC contents of the libraries ranged from 46.02 to 52.94%. All the samples had at least 91.04% reads equal to or exceeding Q30. The above results indicated that the data from these samples were qualified for different expression genes (DEGs) analysis requirement.

### 3.4. Analysis of DEGs at Different Development Stages under Hypotonic Stress of M. rosenbergi Post-Larvae

Under hypotonic stress, the number of DEGs in 7 dph, 14 dph, 21 dph larvae group was shown in Figure 2A–C. At 7 dph, 14 dph and 21 dph larvae, there were 1428, 1187, 132 DEGs including 301, 366, 4 up-regulated genes and 1127, 821, 128 down-regulated genes, respectively (|log FC| ≥ 1, FDR < 0.05). As the developmental stage progresses, the number of DEGs in 7 dph, 14 dph and 21 dph larvae gradually decreased (Figure 2G). Different expression patterns at different development stages were shown in Figure 2D–F. A Venn diagram was generated to visually compare the expression of DEGs (Figure 2H). There were 258, 27, 9 DEGs shared in the compared groups S15-7_S6-7 vs. S15-14_S6-14; S15-14_S6-14 vs. S15-21_S6-21; and S15-7_S6-7 vs. S15-21_S6-21, respectively. The principal component analysis (PCA) score plot displayed that the distance between S15-7 and S6-7, S15-14 and S6-14 was relatively large, and the distance between S15-21 and S6-21 was relatively close. Besides, the distance between S15-7 and S15-14, was relatively close, and far from S15-21. The distance between S6-7, S6-14, and S6-21 was relatively large (Figure 2I).

### 3.5. Functional Analysis by GO Enrichment

In order to further understand the effects of hypotonic stress on biological function at different development stages of *M. rosenbergii* post-larvae, we performed GO enrichment analysis on DEGs of 7 dph, 14 dph, 21 dph larvae under hypotonic stress, mainly including three categories: molecular function (MF), cellular components (CC), and biological process (BP) (Figure 3A–C). At 7dph larvae, the DEGs were enriched mainly in biological regulation, cellular process, metabolic process, and multicellular organismal process in the BP categories; the cell, cell part, and organelle contained the most DEGs in CC categories and catalytic activity and binding were the most abundant functions in MF categories (Figure 3A). At 14 dph larvae, the DEGs were enriched mainly in cell, cell part, and membrane in CC categories, and the most abundant functions in BP and MF categories were the same with 7 dph larvae (Figure 3B). At 21 dph larvae, the number of GO terms enriched for DEGs decreased, and in BP categories, the developmental process and multicellular organismal process were the main GO term (Figure 3C).

### 3.6. Functional Analysis by KEGG Enrichment

In order to further clarify the function of DEGs in signaling pathways, the differently expressed unigenes were located in the KEGG database to analyze significantly enriched pathways at different development stages of *M. rosenbergii* post-larvae (Figure 4). At 7 dph larvae, results showed that the DEGs enriched in KEGG pathways were involved in cellular processes, environmental information processing, human diseases, metabolism, and organismal systems under hypotonic stress (Figure 4A). The significantly enriched KEGG pathways’ results indicated that enriched pathways played an important role in the immune system, such as lysosome, autophagy-animal and antigen processing, and presentation in the stage of 7 dph under hypotonic stress (Figure 4D). At 14 dph larvae, the DEGs enriched in the KEGG pathways were involved in cellular processes, environmental information processing, genetic information processing, human diseases, metabolism, and organismal systems under hypotonic stress (Figure 4B). The significantly enriched KEGG pathways’ results indicated that enriched pathways played an important role in the immune system and endocrine system, such as autophagy-animal and antigen processing and presentation and the renin-angiotensin system at the stage of 14 dph under hypotonic stress (Figure 4E). At 21 dph larvae, the DEGs enriched in KEGG pathways were only involved in cellular processes under hypotonic stress (Figure 4C). The number of significantly enriched KEGG pathways was decreased, and results indicated that the enriched pathways played an important role in endocrine system, such as renin-angiotensin system in the stage of 21 dph under hypotonic stress (Figure 4F).

### 3.7. Data Validation by qPCR

According to the functional analysis by GO and KEGG enrichment pathways above, the autophagy-related genes were selected for qPCR to verify and validate their expression levels in the present study. After clustering autophagy-related genes into two groups with a single heatmap approach (Figure 5A), the relationship with autophagy-related genes between the control and hypotonic stress can be easily detected by visualizing the pattern difference. It was clear that the expression of the beclin 1-associated autophagy-related key regulator (*Barkor*); ubiquitin-like modifier-activating enzyme ATG7 (*ATG7*); *Beclin*; autophagy-related protein 13 (*ATG13*); nuclear receptor-binding factor 2 (*Nrbf2*); ubiquitin-like-conjugating enzyme ATG3 (*ATG3*); vacuole membrane protein 1 (*VMP1*); and autophagy-related protein 2 (*ATG2*) decreased at 7 dph, 14 dph larvae, and then increased at 21 dph larvae under hypotonic stress (Figure 5A). Similarly, the results of the qPCR also showed that the expression of *Barkor; ATG7; Beclin; ATG13; Nrbf2; ATG3; VMP1;* and *ATG2* decreased at 7 dph, 14 dph larvae, and then increased at 21 dph larvae under hypotonic stress (Figure 5B). We found that the upregulation and downregulation trends of qPCR results are similar to the RNA-Seq results, indicating that the transcriptome sequencing data are reliable.

### 3.8. Correlation between Antioxidant Parameters and DEGs

To further explore the potential relationship between antioxidant parameters and the autophagy-related genes, heatmap visualization was performed (Figure 6A–D). There were negative correlations between MDA and all of the genes, while GSH-Px activity and NO level had a significant positive relationship with all of the genes in hypotonic stress group (Figure 6A). In the 7 dph larvae hypotonic stress group, the MDA level had a significant negative relationship with *ATG13*, while ASAFR had a positive relationship with *ATG13*. The iNOS level had a significant negative relationship with *ATG2* (Figure 6B). In 14 dph larvae hypotonic stress group, the GSH-Px had a positive relationship with *Nrbf2*, the ASAFR was positive with *ATG7* while negative with *Barkor*, and the MDA had a negative relationship with *Beclin* (Figure 6C). In the 21 dph larvae hypotonic stress group, the ASAFR was positive with *ATG3* while negative with *VMP1*, the MDA level was negative with *Beclin*, and the iNOS level was positive with *ATG13* (Figure 6D).

## 4. Discussion

A previous study discovered that larvae of *M. rosenbergii* required suitable salinity for survival and growth [28]. Several studies reported the effects of salinity on rates of protein synthesis, oxygen uptake, growth, survival, and proximate composition in the juveniles of *M. rosenbergii* [29,30,31]. However, all the above studies concentrated on the juvenile stage, not the post-larvae stage of *M. rosenbergii*. This work set out to evaluate the effects of hypotonic stress on the antioxidant system and underlying mechanisms at different development stages (7 dph, 14 dph, 21 dph larvae) of the *M. rosenbergii* post-larvae. 

The organisms developed antioxidant defense mechanisms to reduce oxidative stress and protect the biological systems from free radical toxicity [21,31]. The primary antioxidant defense system included activated antioxidant enzymes that scavenged the reactive oxygen species, such as SOD and GSH-Px [32]. SOD could scavenge reactive oxygen species (ROS) and reduce lipid peroxidation injury, and many studies reported the importance of SOD in protecting cells against oxidative stress [33,34]. GSH-Px is an essential element of the antioxidative system in cells, which is the catalysis for the conversion of hydrogen peroxide and fatty acid hydro-peroxides into water and fatty acid alcohol, thereby protecting cell membranes against oxidative damage [35]. ASAFR was the body’s ability to scavenge the superoxide anion ROS, and numerous studies have shown the importance of ASAFR in protecting cells against oxidative stress [36,37]. Previous studies showed that salinity stress decreased SOD activity in freshwater fish (*Oreochromis niloticus*) [38] and juvenile *Pampus argenteus* [39]. Meanwhile, several studies reported that salinity stress reduced GSH-Px activities in freshwater fish (*Oreochromis niloticus*) [38] and in the juvenile silver pomfret, *Pampus argenteus* [40]. Besides, a previous study reported that the ASAFR was significantly reduced under stress in juvenile *Megalobrama amblycephala* [41]. Similar with above studies in our own results, the activity of SOD, GSH-Px, and ASAFR concentrations were remarkably decreased by low salinity in the S6 group at different developmental stages of post-larvae, which might imply that hypotonic stress reduced the antioxidant capacity of post-larvae. Besides, the level of SOD, GSH-Px, and ASAFR concentrations increased significantly with the developmental stages of 7 dph, 14 dph and 21 dph in the S15 and S6 groups. It indicated that the antioxidant capability was enhanced with the developmental stages in *M. rosenbergii* post-larvae. Similar results have been reported, showing that the activities of SOD and GSH-Px were enhanced with age in sea animals and *Acipenser naccarii* [42,43]. These studies further confirmed our conjecture. Interestingly, the activity of SOD, GSH-Px, and ASAFR concentration in the S6 group were also increased with the developmental stages and was significantly lower than the S15 group, which may be due to the post-larvae gradually adapting to hypotonic stress with the developmental stages. However, the specific adaptation mechanism needs to be further studied.

MDA is a byproduct of membrane lipid peroxidation and is frequently used as a marker of cell membrane damage caused by oxidative stress [44]. Previous studies showed that low salinity stimulation increased MDA levels in the European sea bass (*Dicentrarchus labrax*) [45] and enhanced the production of lipid peroxidation in sturgeons (*Acipenser naccarii*) [46]. Similar to previous studies, our results exhibited that MDA levels significantly increased under hypotonic stress stimulation in the 7 dph larvae, while it showed no significant difference in the 14 dph and 21 dph larvae. This may indicate that generation of ROS was induced by the hyposaline environment, eventually inducing oxidative damage. In addition, the MDA level decreased with the developmental stages in the present study. Previous research in the antioxidant activity and MDA levels in adult and embryonic zebrafish showed that the adults were more tolerant to environmental stress than the embryonic zebrafish [47]. It might imply that the tolerance of post-larvae to hypotonic stress gradually enhanced with the increase in developmental stages. 

The iNOS-derived NO signaling plays a central role in the regulation of several biochemical pathways during inflammatory conditions [48,49]. The iNOS-NO signaling improved the antimicrobial activity of *Litopenaeus vannamei* and *Fenneropenaeus chinensis* [50,51]. A previous study showed that the iNOS was activated and induced NO release in injured liver [52], and the level of NO was increased significantly under *Vibrio anguillarum* and ammonia stress in *M. rosenbergii* [53]. In the present study, the levels of NO and iNOS were increased in the hypotonic stress group in post-larvae, which might indicate hypotonic stress caused free radical injury in post-larvae. 

To obtain a more comprehensive understanding of molecular regulatory mechanisms of hypotonic stress at different development stages, the transcriptomic analysis was performed at different development stages of *M. rosenbergii* post-larvae. A total of 1428, 1187, 132 DEGs were identified in the hypotonic stress group of 7 dph, 14 dph, 21 dph larvae, respectively. These DEGs were subject to further analysis through a series of bioinformatic methods that sought to understand the molecular mechanisms that were affected by hypotonic stress at different development stages of *M. rosenbergii* post-larvae. The PCA analysis showed that the gene expression patterns in the 7 dph and 14 dph larvae between the S15 and S6 groups were different while showed a similar pattern in the 21 dph larvae. Previous studies showed that the different developmental stages of *Litopenaeus vannamei* showed different levels of adaptability to low salinity [54,55]. The results of the present study indicated that a large number of gene expressions changed in response to the hypotonic stress in the early stages of post-larvae, while this change gradually weakened with the developmental stages of post-larvae. We speculated that it might be due to the increase in the adaptability of post-larvae with the developmental stage of hypotonic stress. 

We performed GO and KEGG enrichment analysis of DEGs in order to further explore the research results. Our results showed that the number of GO enrichment terms and KEGG pathways was decreased with the post-larvae developmental stages. It might imply that the response to hypotonic stress in post-larvae gradually reduced. Previous studies reported that the binding could maintain osmotic pressure homeostasis in the euryhaline teleost *Fundulus heteroclitus*, and the catalytic function of enzymes regulated ion changes and osmotic pressure when crustaceans were exposed to salinity fluctuations [56,57]. In a recent study, GO enrichment analysis showed that DEGs were significantly enriched in the binding and catalytic activities of molecular function, which might indicate that the function of regulated ion changes and osmotic pressure was activated in post-larvae under hypotonic stress. 

Salinity stress can cause changes in the enrichment of some pathways. Under hypotonic stress, the results of KEGG analysis showed that the lysosome, antigen processing and presentation, and autophagy pathways were significantly enriched in the 7 dph larvae, and lysosome, antigen processing and presentation, and renin-angiotensin system (RAS) pathways were significantly enriched in the 14 dph larvae. However, in the 21 dph larvae, the only significantly enriched pathway was RAS. Interestingly, the autophagy pathway only enriched significantly in the early hypotonic stress stage (7 dph), while it was not significantly enriched in the later stages (14 and 21 dph). Autophagy is a relatively conservative cell degradation process in evolution, through which cells recycle ineffectual cellular components (such as damaged DNA, damaged organelles, intracellular pathogens, and misfolded proteins) to maintain homeostasis [56]. The autophagy-lysosome system could clean the oxidized, damaged, and misfolded proteins induced by environmental stress in the cell [58,59]. A recent study reported that low temperature-induced autophagy is a strategy to adapt to and cope with stress in *Penaeus vannamei* [59]. However, previous research showed that the autophagy pathway was activated under mild oxidative stress in order to initiate cell survival and repair mechanisms, but severe oxidative stress would impair autophagy process, accelerating cell apoptosis, and necrosis [60]. The occurrence of autophagy is usually accompanied by the activation of some autophagy factors. Beclin 1 is an important autophagy effector, involved in phagophore nucleation (early autophagosome formation) [61]. At the same time, the ubiquitin-like proteins like Atg7, Atg3, and Atg8 were required in autophagosome biogenesis [62]. In this study, the autophagy-related genes such as *beclin1*, *atg3*, *atg7* or *atg13* were obviously downregulated in the 7 dph larvae, while upregulated in the 21 dph larvae under hypotonic stress. We speculated that hypotonic stress caused an impaired autophagic process which might accelerate cell apoptosis and cause post-larvae damage in the early stage of post-larvae, while the osmoregulatory capacity gradually made the post-larvae adapt to hypotonic stress and the impaired autophagic process was gradually relieved. It was similar to the change patterns of the anti-oxidative enzyme systems.

A previous study reported that the osmoregulatory capacity was correlated with the developmental stage [17], which was consistent with our point of view. Besides, the results of correlation between antioxidant parameters and autophagy related genes showed that GSH-Px, MDA, NO were significant correlated with the expression of autophagy related genes. The essential autophagy effector Beclin was negative significantly with MDA in 14, 21 dph larvae under hypotonic stress, while showed no significant correlation in 7 dph larvae. It indicated that the oxidative damage induced by hypotonic stress was correlated with the expression of Beclin. However, the specific related mechanisms need to be further studied. In the present study, renin-angiotensin system (RAS) was activated at 14 dph and 21 dph larvae under hypotonic stress. RAS is involved in the regulation of drinking rates and blood pressure, therefore to guarantee fish survival in media with different osmolarities [63]. It was suggested that the response to hypotonic stress in the 14 dph and 21 dph larvae was mainly through RAS. However, the mechanism of salinity adaptability with the developmental stages needs further study.

## 5. Conclusions

In the present study, our results demonstrated that hypotonic stress reduced antioxidant and immune capacity at different development stages of *M. rosenbergii* post-larvae. Further, we used transcriptome analysis to evaluate the underlying molecular mechanisms of hypotonic stress in *M. rosenbergii* post-larvae. The potential mechanism of hypotonic stress is summarized in Figure 7. Results showed that the number of DEGs decreased with the developmental stages in *M. rosenbergii* post-larvae under hypotonic stress. GO and KEGG enrichment revealed that autophagy dysregulated in the 7 dph and14 dph larvae under hypotonic stress. The autophagy related genes, including *Barkor*, *ATG13*, *ATG7*, *Beclin*, *Nrbf2*, *ATG3*, *VMP1,* and *ATG2,* decreased in the7 dph, 14 dph larvae, and then increased in the 21 dph larvae under hypotonic stress. With the developmental stages of post-larvae, the RAS became a significantly enriched signaling pathway under hypotonic stress. In conclusion, hypotonic stress reduced the antioxidant capacity and impaired the immune system in post-larvae, but as development progresses, the adaptability of post-larvae to hypotonic stress gradually increased, and might reach a new homeostasis through the RAS signaling pathway. This study provided novel insight into the underlying mechanism of hypotonic stress at different development stages of *M. rosenbergii* post-larvae, and provides an extensive resource for the future culture development of *M. rosenbergii*.

## Figures and Tables

**Figure 1 antioxidants-11-00440-f001:**
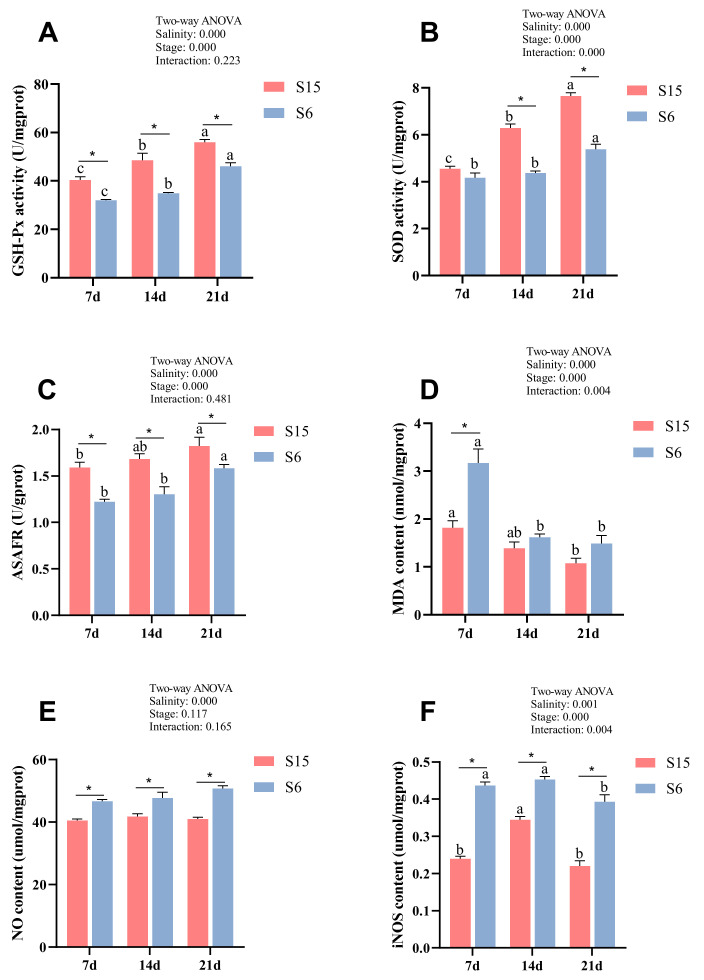
Combined effect of salinity and different development stages on Glutathione peroxidase (**A**, GSH-Px); superoxide dismutase (**B**, SOD); anti-superoxide anion free radical compound (**C**, ASAFR); malondialdehyde (**D**, MDA); nitric oxide (**E**, NO); inducible nitric oxide synthase (**F**, iNOS) of *Macrobrachium rosenbergii* post-larvae. Note: Data are mean values of nine replicates expressed as mean ± SE. “a, b, c” showed significant differences (*p* < 0.05) in the same group between different development stages. “*” showed significant differences (*p* < 0.05) between different salinity under the same development stages.

**Figure 2 antioxidants-11-00440-f002:**
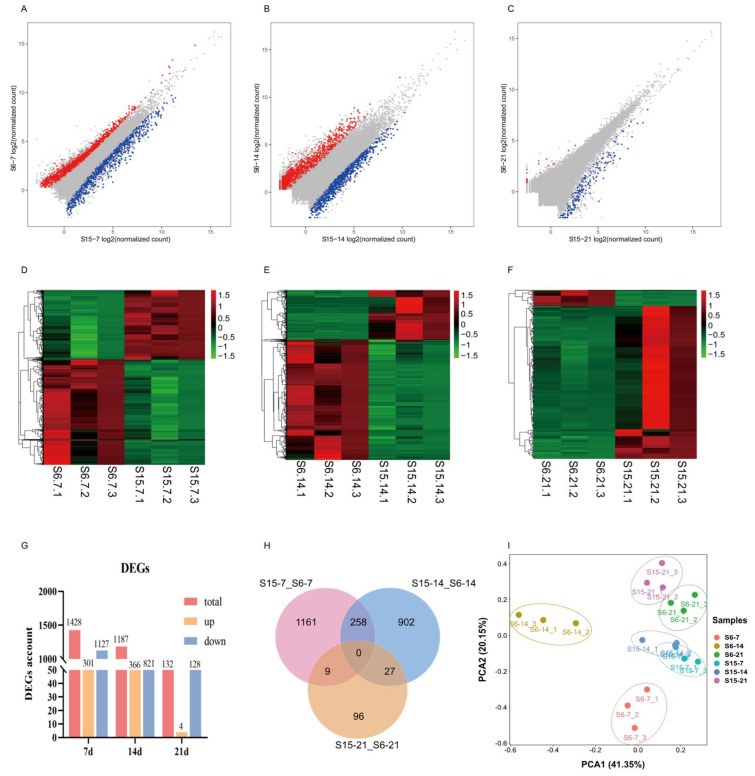
Effects of hypotonic stress at different development stages of *M. rosenbergii* post-larvae on gene expression. Note: (**A**–**C**): the scatter plots showed the differences of gene expression of hypotonic stress in 7, 14, 21 day-post-hatch larvae, respectively; Red represents upregulated genes, blue represents downregulated genes; (**D**–**F**): the heatmap showed differential gene expression of hypotonic stress in 7, 14, 21 day-post-hatch larvae, respectively; Red represents upregulated genes, green represents downregulated genes; (**G**): the histogram showed the counts of different expression genes in different group; (**H**): Venn diagram comparing the expression distribution of different expression genes; (**I**): Principal component analysis (PCA) for the correlation among samples in different group. S6-7: 7 day-post-hatch larvae in 6 psu salinity; S15-7: 7 day-post-hatch larvae in 15 psu salinity; S6-14: 14 day-post-hatch larvae in 6 psu salinity; S15-14: 14 day-post-hatch larvae in 15 psu salinity; S6-21: 21 day-post-hatch larvae in 6 psu salinity; S15-21: 21 day-post-hatch larvae in 15 psu salinity.

**Figure 3 antioxidants-11-00440-f003:**
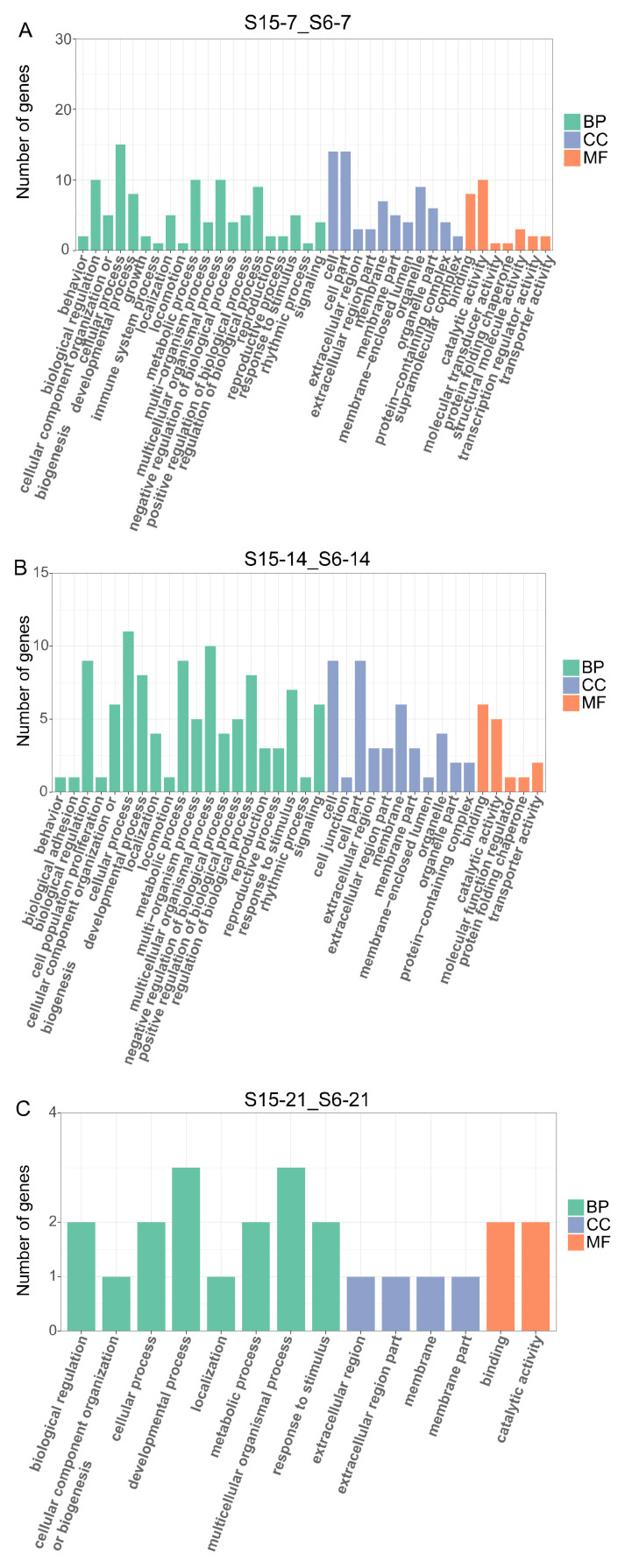
Enriched GO terms for DEGs in *M.rosenbergii* post-larvae under hypotonic stress at different development stages. Note: (**A**): DEGs of comparison group S15-7_S6-7 enriched GO terms; (**B**): DEGs of comparison group S15-14_S6-14 enriched GO terms; (**C**): DEGs of comparison group S15-21_S6-21enriched GO terms. S6-7: 7 day-post-hatch larvae in 6 psu salinity; S15-7: 7 day-post-hatch larvae in 15 psu salinity; S6-14: 14 day-post-hatch larvae in 6 psu salinity; S15-14: 14 day-post-hatch larvae in 15 psu salinity; S6-21: 21 day-post-hatch larvae in 6 psu salinity; S15-21: 21 day-post-hatch larvae in 15 psu salinity.

**Figure 4 antioxidants-11-00440-f004:**
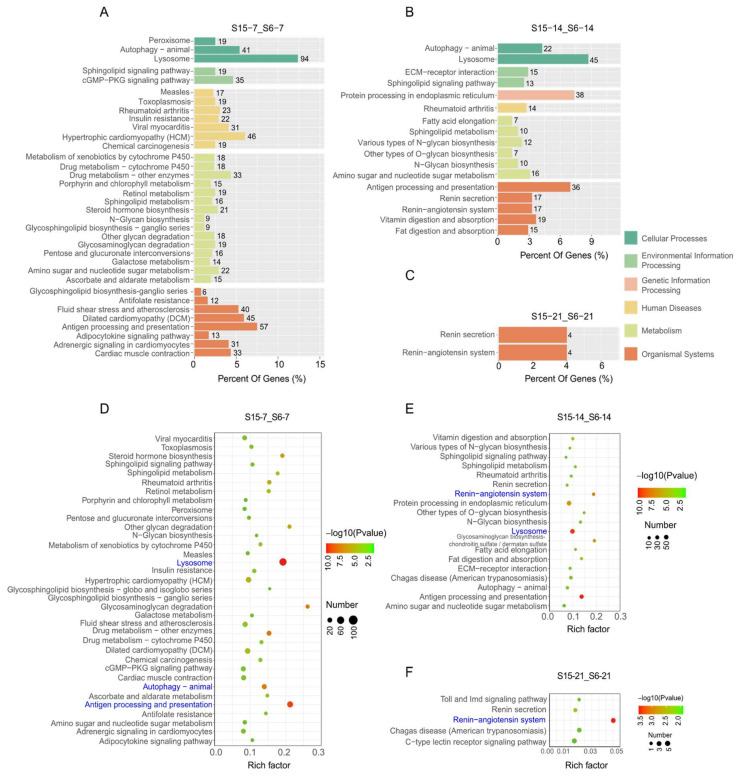
Enriched KEGG pathways for DEGs in *M.rosenbergii* post-larvae under hypotonic stress at different development stages. Note: (**A**–**C**): KEGG pathways involved in different categories at 7 dph, 14 dph, 21 dph larvae; (**D**–**F**): The most significant enrichment KEGG pathways at 7 dph, 14 dph, 21 dph larvae. S6-7: 7 day-post-hatch larvae in 6 psu salinity; S15-7: 7 day-post-hatch larvae in 15 psu salinity; S6-14: 14 day-post-hatch larvae in 6 psu salinity; S15-14: 14 day-post-hatch larvae in 15 psu salinity; S6-21: 21 day-post-hatch larvae in 6 psu salinity; S15-21: 21 day-post-hatch larvae in 15 psu salinity.

**Figure 5 antioxidants-11-00440-f005:**
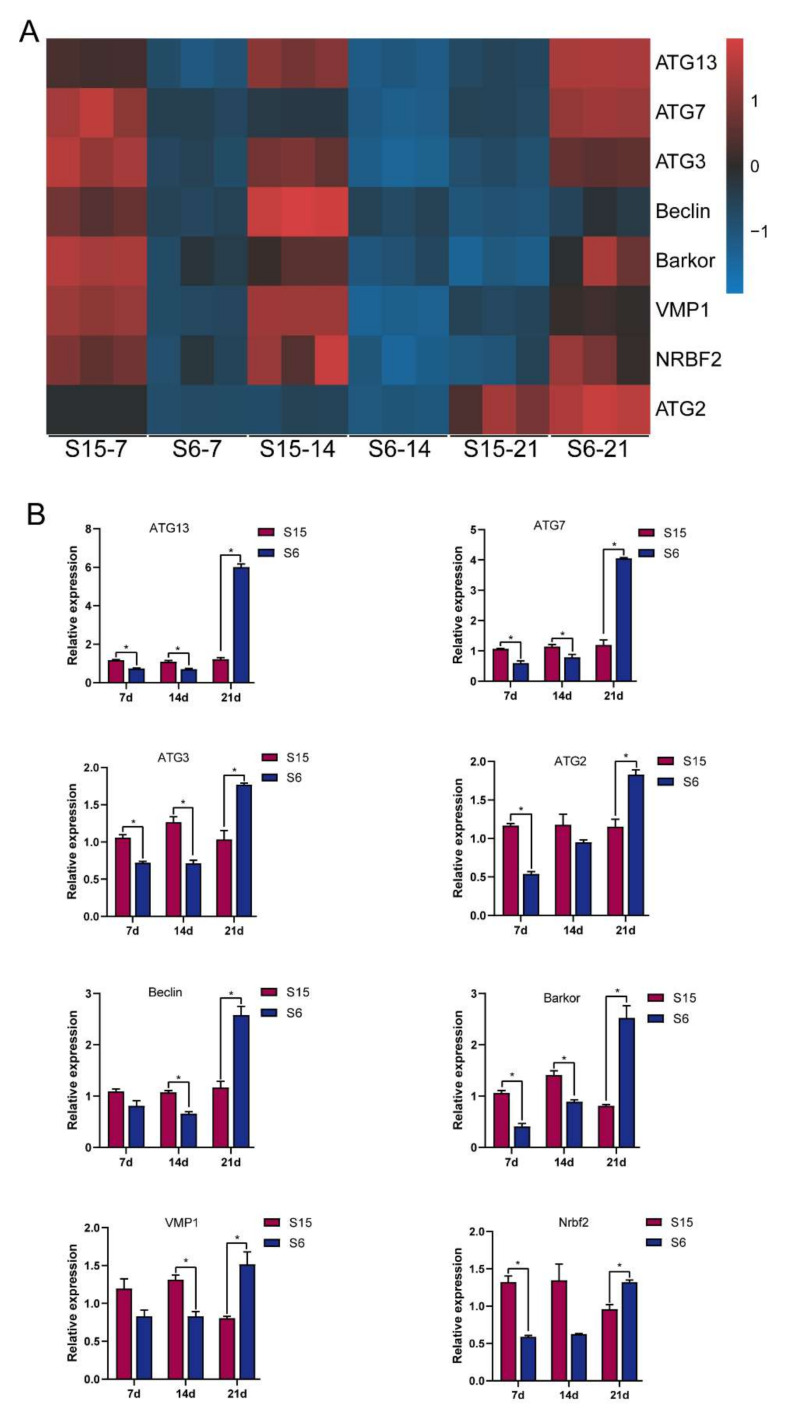
Relative levels of autophagy signaling based on transcriptome analysis and qPCR validated. Note: (**A**): heatmap of relative levels of key genes of autophagy signaling retrieved from RNA-seq database; (**B**): relative transcripts levels (qPCR validation) of autophagy signaling key genes, β-actin was used as an internal control. * *p* < 0.05.

**Figure 6 antioxidants-11-00440-f006:**
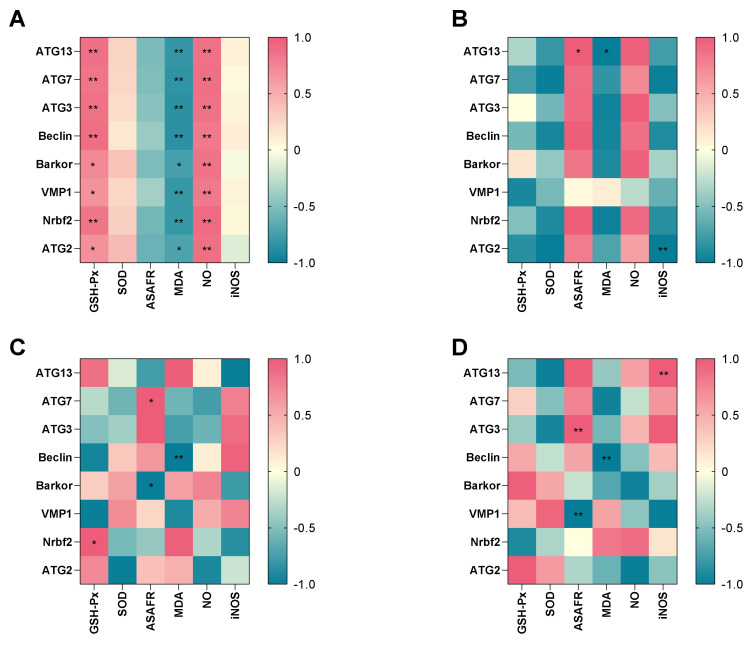
Correlation between antioxidant parameters and key different expression genes in hypotonic stress group. Note: (**A**): the correlation in hypotonic stress group; (**B**): the correlation in 7 dph larvae under hypotonic stress; (**C**): the correlation in 14 dph larvae under hypotonic stress; (**D**): the correlation in 21 dph larvae under hypotonic stress. The heat-map shows the Pearson correlation coefficient. The intensity of the colors represents the degree of association. * *p* < 0.05, ** *p* < 0.01.

**Figure 7 antioxidants-11-00440-f007:**
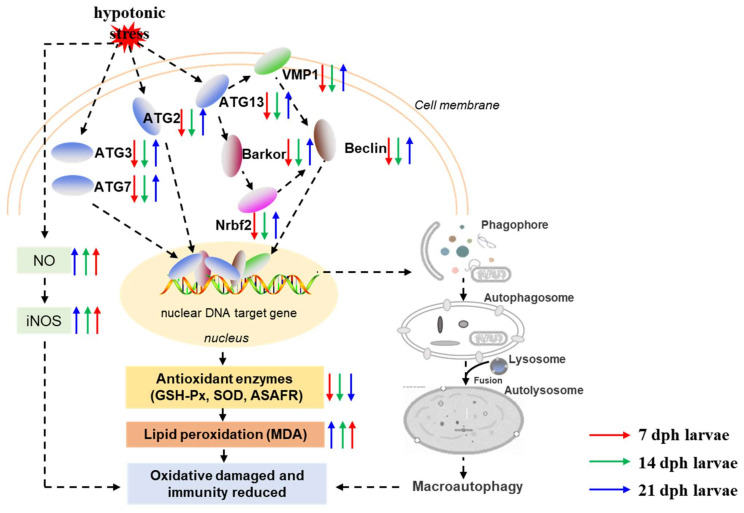
Schematic diagram depicting the effects and possible mechanisms of hypotonic stress on antioxidant, immune capacity and autophagy related genes expression. Notes: The red arrow is representative of 7 dph larvae, green arrow is representative of 14 dph larvae, and blue arrow is representative of 21 dph larvae under hypotonic stress compare with normal salinity group; The upward arrow is representative of gene expression activation and the downward arrow is inhibition of gene expression.

## Data Availability

Data is contained within the article.

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
