# Peer review of "Application of Transcriptome Analysis to Understand the Adverse Effects of Hypotonic Stress on Different Development Stages in the Giant Freshwater Prawn Macrobrachium rosenbergii Post-Larvae"

_antioxidants, 2022, doi:10.3390/antiox11030440_

Round 1

Reviewer 1 Report

Reviewer Report

This is a very exciting study that investigated the application of transcriptome analysis to understand the adverse effects of hypotonic stress on different development stages in giant freshwater prawn Macrobrachium rosenbergi post-larvae. The authors found that reducing salinity at the post-larvae stage causes oxidative stress as shown in their results with decreased in gluthathione peroxidase, superoxide dismutase and increased in oxidative damage product MDA. They also found that hypotonic stress causes impairment of immune response machineries. The manuscript would be a good fit for the journal and advances our understanding of the stress induced by low salinity during post-larvae development of giant freshwater prawn Macrobrachium rosenbergi. I have the following major concerns about the manuscript in it current form that the authors may consider.

Major concerns:

  1. In this manuscript the authors used three different salinity units- psu, S, and % The standard unit for salinity is psu or ppt. While some journals suggest reporting just the number e.g 15 or 6 without the unit. I suggest using psu throughout the manuscript.
  2. It is not clear the duration of the post-larvae stage development in the life cycle of giant freshwater prawn Macrobrachium rosenbergi. This window is important for this manuscript, it should be clear in the introduction and forming the rational for the variation of salinity.
  3. The authors mentioned salinity 2 to 40 psu what is the rational for using 6 psu and 15 psu
  4. The experimental design as written is confusing my understanding is that following post-larvae stage- first group were divided into two salinity 15 and 6 psu and kept for 21 days. The second group were sampled at different days post-larvae, 7, 14 and 21 at different salinities. It is not clear to me whether the salinity treatments were acute or chronic. What is S15-7, S6-7? Is it 15 to 7 salinity acutely or chronic for 7 days and 6 to 7 salinity? This need clear explanation for readers to understand the study.

Reviewer 2 Report

The manuscript n. antioxidants-1592171 reports new and very interesting scientific data on physiological responses of Macrobrachium rosenbergii post-larvae exposed to hypotonic stress.

The manuscript is well organized, the methodological approach is in general correct and adequate to the research. The discussion well argued. I consider that this manuscript will be appropriate for publication in Antioxidants after some revisions.

1) Has the purified RNA been evaluated for purity? The 260/280 and 260/230 absorbance ratios are usually applied.

2) The Authors should specify the cDNA synthesis and amplification methods, also detailing the thermal cycles.

3) Lines 356-357. “Organisms developed antioxidant defense mechanisms to reduce oxidative stress and protect biological systems from free radical toxicity [31].” The reference 31 is focused on only the common carp. I suggest the Authors to add a couple of other references focused on other organism.

4) Lines 357, 359 and 405. Please refer to “ROS” instead of “free radicals”.

5) The Author only considered SOD and GPX as antioxidant enzymes. How come? In any case, in my opinion it would be necessary to discuss the fact that other antioxidant enzymes such as catalase and peroxiredoxins can also play an important role in protecting the organism against oxidative stress induced by hypotonicity, also presenting it as a future target.

6) I found a few typos in the text that have to be fixed. For example, M. rosenbergii shold be written in italics also in the abstract (there are other species names to correct also in the reference list), and the Y axis font of figure 1A should be the same of other figures.

Round 2

Reviewer 1 Report

The authors have addressed all my concerns.